# [Re] Don't Judge an Object by Its Context: Learning to Overcome Contextual Bias

**Sunnie S. Y. Kim   Sharon Zhang   Nicole Meister   Olga Russakovsky**
Princeton University
{sunniesuhyoung, sharonz, nmeister, olgarus}@princeton.edu

## Reproducibility Summary

**Scope of Reproducibility**

Singh et al. [9] point out the dangers of *contextual bias* in visual recognition datasets. They propose two methods, *CAM-based* and *feature-split*, that better recognize an object or attribute in the absence of its typical context while maintaining competitive within-context accuracy. To verify their performance, we attempted to reproduce all 12 tables in the original paper, including those in the appendix. We also conducted additional experiments to better understand the proposed methods, including increasing the regularization in *CAM-based* and removing the weighted loss in *feature-split*.

**Methodology**

As the original code was not made available, we implemented the entire pipeline from scratch in PyTorch 1.7.0. Our implementation is based on the paper and email exchanges with the authors. [1] We spent approximately four months reproducing the paper, with the first two months focused on implementing all 10 methods studied in the paper and the next two months focused on reproducing the experiments in the paper and refining our implementation. Total training times for each method ranged from 35–43 hours on COCO-Stuff [1], 22–29 hours on DeepFashion [7], and 7–8 hours on Animals with Attributes [10] on a single RTX 3090 GPU.

**Results**

We found that both proposed methods in the original paper help mitigate contextual bias, although for some methods, we could not completely replicate the quantitative results in the paper even after completing an extensive hyperparameter search. For example, on COCO-Stuff, DeepFashion, and UnRel, our *feature-split* model achieved an increase in accuracy on out-of-context images over the standard baseline, whereas on AwA, we saw a drop in performance. For the proposed *CAM-based* method, we were able to reproduce the original paper's results to within 0.5% mAP.

**What was easy**

Overall, it was easy to follow the explanation and reasoning of the experiments. The implementation of most (7 of 10) methods was straightforward, especially after we received additional details from the original authors.

**What was difficult**

Since there was no existing code, we spent considerable time and effort re-implementing the entire pipeline from scratch and making sure that most, if not all, training/evaluation details are true to the experiments in the paper. For several methods, we went through many iterations of experiments until we were confident that our implementation was accurate.

**Communication with original authors**

We reached out to the authors several times via email to ask for clarifications and additional implementation details. The authors were very responsive to our questions, and we are extremely grateful for their detailed and timely responses.

---

[1]Our implementation can be found at `https://github.com/princetonvisualai/ContextualBias`.

# 1 Introduction

Most prominent vision datasets are afflicted by *contextual bias*. For example, "microwave" typically is found in kitchens, which also contain objects like "refrigerator" and "oven." Such co-occurrence patterns may inadvertently induce contextual bias in datasets, which could consequently seep into models trained on them. When models overly rely on context, they may not generalize to settings where typical co-occurrence patterns are absent. The original paper by Singh et al. [9] proposes two methods for mitigating such contextual biases and improving the robustness of the learnt feature representations. The paper demonstrates their methods on multi-label object and attribute classification tasks, using the COCO-Stuff [1], DeepFashion [7], Animals with Attributes (AwA) [10], and UnRel [8] datasets. Our exploration centers on four main directions:

First, we trained the baseline classifier presented in the paper (Section 2.1 for implementation; Sections 2.3-2.4 for results). Due to likely implementation discrepancies, our results differed from the original paper by 0.6–3.1% mAP on COCO-Stuff, by 0.7–1.4% top-3 recall on DeepFashion, and by 0.1–3.2% mAP on AwA (Table 2). We ran a hyperparameter search (Appendix C), which yielded a significant (1.4–3.6%) improvement on DeepFashion.

Next, we identified the *biased categories* in each dataset, i.e., visual categories that suffer from contextual bias. We followed the proposed method of using the baseline classifier to identify these categories, and discovered that the classifier implementation has a non-trivial effect. For COCO-Stuff, 18 of the top-20 categories we identified matched the original paper's top-20 categories (10 on DeepFashion, 18 on AwA; Section 2.2). Nevertheless, the categories we identified appear reasonable (e.g., "fork" co-occurs with "dining table"; Appendix B). As training and evaluation of most methods depend on the biased categories, we used the paper's biased categories for subsequent experiments.

Third, we checked the main claim of the paper, that the proposed *CAM-based* and *feature-split* methods help improve recognition of biased categories in the absence of their context (Section 3). On COCO-Stuff, DeepFashion, and UnRel, we were able to reproduce the improvements gained from the proposed *feature-split* method towards reducing contextual bias, whereas on AwA, we saw a drop in performance. The proposed *CAM-based* method, which was only applied to COCO-Stuff, also helped reduce contextual bias, though not as significantly as the *feature-split* method. For this method, we reproduced the original paper's results to within 0.5% mAP (Section 3.5). We also successfully reproduced the paper's weight similarity analysis, as well as the qualitative analyses on class activation maps (CAMs) [12].

Lastly, we ran additional experiments and ablation studies (Section 3.6). These revealed that the regularization term in the *CAM-based* method and the weighted loss in the *feature-split* method are central to the methods' performance. We also observed that varying the feature subspace size influences the *feature-split* method accuracy.

# 2 Reproducing the *standard* baseline and the biased category pairs

The first step in reproducing the original paper is doing "stage 1" training. This stage involves training a *standard* multi-label classifier with the binary cross entropy loss on the COCO-Stuff, DeepFashion, and AwA datasets. We describe how we obtained and processed the datasets in Appendix A. The *standard* model is used to identify the biased categories and serves as a starting point for all "stage 2" methods, i.e., the proposed *CAM-based* and *feature-split* methods and 7 other strong baselines introduced in Section 3.

## 2.1 Implementation and training details

According to the original paper, all models use ResNet-50 [4] pre-trained on ImageNet [3] as a backbone and are optimized with stochastic gradient descent (SGD) and a batch size of 200. Each *standard* model is optimized with an initial learning rate of 0.1, later dropped to 0.01 following a standard step decay process. The input images are randomly resize-cropped to 224×224 and randomly flipped horizontally during training. We also received additional details from the authors that SGD is used with a momentum of 0.9 and no weight decay. The COCO-Stuff *standard* model is trained for 100 epochs with the learning rate reduced from 0.1 to 0.01 after epoch 60. The DeepFashion *standard* model is trained for 50 epochs with the learning rate reduced after epoch 30. The AwA *standard* model is trained for 20 epochs with the learning rate reduced after epoch 10.

After training with the paper's hyperparameters, we found that our reproduced *standard* models for COCO-Stuff and AwA were consistently underperforming against the results in the paper. Thus, we also tried varying the learning rate, weight decay, and the epoch at which the learning rate is dropped to achieve the best possible results. Further details can be found in Appendix C. On both COCO-Stuff and AwA, our hyperparameter search ended up reconfirming the original paper's hyperparameters as the optimal ones; for DeepFashion, we were able to find an improvement. The original, reproduced and tuned results are shown in Table 1, following explanations of biased categories identification (Section 2.2) and evaluation details (Section 2.3).

## 2.2 Biased categories identification

The paper identifies the top-20 $(b, c)$ pairs of biased categories for each dataset, where $b$ is the category suffering from contextual bias and $c$ is the associated context category. This identification is crucial as it concretely defines the contextual bias the paper aims to tackle, and influences the training of the "stage 2" models and evaluation of all models.

The paper defines *bias* between two categories $b$ and $z$ as the ratio between average prediction probabilities of $b$ when it occurs with and without $z$. Note that this definition of bias requires a trained model, unlike the more common definition of bias that only requires co-occurrence counts in a dataset [11]. Following the paper description, we used a *standard* model trained on an 80-20 split for COCO-Stuff and one trained on the full training set for DeepFashion and AwA. For each category $b$ in a given dataset, we calculated the bias between $b$ and its frequently co-occuring categories, and defined category $c$ as the context category that most biases $b$, i.e. has the highest bias value. Bias is calculated on the 20 split for COCO-Stuff, the validation set for DeepFashion, and the test set for AwA.[2] After the bias calculation, we identified 20 $(b, c)$ pairs with the highest bias values. The paper emphasizes that this definition of bias is directional; it only captures the bias $c$ incurs on $b$ and not the other way around.

We compare our pairs to the paper's in Tables A1 (COCO-Stuff), A2 (DeepFashion), and A3 (AwA) in the Appendix. Out of 20 biased categories, 2 of ours differed from the paper's for COCO-Stuff, 10 differed for DeepFashion, and 2 differed for AwA. The variability is expected, as bias is defined as a ratio of a trained model's average prediction probabilities which will vary across different models. Nonetheless, we found our pairs to also be reasonable, as our biased categories occur frequently with their context categories and rarely without them. See Appendix B for details.

## 2.3 Evaluation details

The paper does not specify image preprocessing or model selection. Following common practice, we resize an image so that its smaller edge is 256 and then apply one of two 224x224 cropping methods: a center-crop or a ten-crop. Both are deterministic procedures. We observed that results with center-crop are consistently better and closer to the paper's results, hence for all experiments, we report results using center-crop. In our email communications, the authors also specified that they use the model at the end of training as the final model. We confirmed that this is a reasonable model selection method after trying three other selection methods, described in Appendix D.

We emphasize that model evaluation is dependent on the identified biased category pairs. For each $(b, c)$ pair, the test set can be divided into three sets: *co-occurring* images that contain both $b$ and $c$, *exclusive* images that contain $b$ but not $c$, and *other* images that do not contain $b$. Then for each $(b, c)$ pair, the paper constructs two test distributions: 1) the "exclusive" distribution containing *exclusive* and *other* images and 2) the "co-occur" distribution containing *co-occurring* and *other* images. We suspect that *other* images are included in both distributions because otherwise, both distributions would have small sizes and only consist of positive images where $b$ occurs, disabling the mAP calculation.

The test distribution sizes can be calculated from the co-occurring and exclusive image counts in Tables A1, A2, A3 in the Appendix. As an example, for the (ski, person) pair in COCO-Stuff, there are 984 co-occuring, 9 exclusive, and 39,511 other images in the test set. Hence, there are $9 + 39,511 = 39,520$ images in the "exclusive" distribution and $984 + 39,511 = 40,495$ images in the "co-occur" distribution. For COCO-Stuff, we also report results on the entire test set (40,504 images) for 60 non-biased object categories and for all 171 categories, following the paper.

---

[2]We received additional information from the original authors that they restricted their COCO-Stuff biased categories to the 80 object categories and performed manual cleaning of the DeepFashion $(b, c)$ pairs.

| Dataset (Metric) | Model | Exclusive | | Co-occur | | Non-biased | | All |
|---|---|---|---|---|---|---|---|---|
| | | Paper BC | Our BC | Paper BC | Our BC | Paper BC | Our BC | |
| COCO-Stuff (mAP) | Paper | **24.5** | - | **66.2** | - | **75.4** | - | **57.2** |
| | Ours (paper params*) | 23.9 | 20.6 | 65.0 | 63.7 | 72.3 | 72.9 | 55.7 |
| DeepFashion (top-3 recall) | Paper | 4.9 | - | 17.8 | - | - | - | - |
| | Ours (paper params) | 5.6 | 5.0 | 19.2 | 15.0 | - | - | - |
| | Ours (tuned params) | **7.0** | 6.3 | **22.8** | 18.4 | - | - | - |
| AwA (mAP) | Paper | 19.4 | - | **72.2** | - | - | - | - |
| | Ours (paper params*) | **19.5** | 21.7 | 69.0 | 69.9 | - | - | - |

Table 1: Reproduced *standard* baseline results on three datasets. We evaluate the models on different subsets of categories/images ("exclusive" and "co-occur" distributions for the 20 biased categories and the entire test set for non-biased and all categories; Section 2.3), both using the paper's and our identified biased category (BC) pairs. *On COCO-Stuff and AwA, hyperparameter tuning did not improve on the original paper's hyperparameters.

For COCO-Stuff and AwA, we calculate the average precision (AP) for each biased category $b$, and report the mean AP (mAP) for each test distribution. For DeepFashion, we calculate the per-category top-3 recall and report the mean value for each test distribution. Higher values indicate a better classifier for both metrics.

## 2.4 Results

In Table 1, we report the original, reproduced, and tuned results with the paper's and our 20 most biased category pairs. Evaluated on the paper's pairs, our best COCO-Stuff model underperforms the paper's by 1-3%, our best DeepFashion model outperforms by 2-5%, and our best AwA underperforms on the "co-occur" distribution by 3.2% and matches the "exclusive" distribution within 0.1%. When we evaluate the same models on our biased category pairs, we get similar results for the AwA model, slightly worse results for the DeepFashion model, and significantly worse results for the COCO-Stuff model. Due to this big drop in performance for COCO-Stuff, which we suspect is caused by the discrepancy in the identified biased category pairs, we choose to use the paper's pairs for training and evaluation in the subsequent sections. Overall, we conclude that the paper's *standard* baseline results are reproducible as we were able to train models within a reasonable margin of error.

## 3 Reproducing the "stage 2" methods: CAM-based, feature-split, and strong baselines

In this section, we describe our efforts in reproducing methods that aim to mitigate contextual bias: namely, the *CAM-based* and *feature-split* methods proposed by the original authors (Figure 1) and 7 other strong baselines. These are referred to as "stage 2" methods because they are trained on top of the "stage 1" *standard* model (except for one strong baseline). Apart from the *feature-split* method, which we discussed with the authors, all other implementations were based entirely on our interpretation of their descriptions in the original paper.

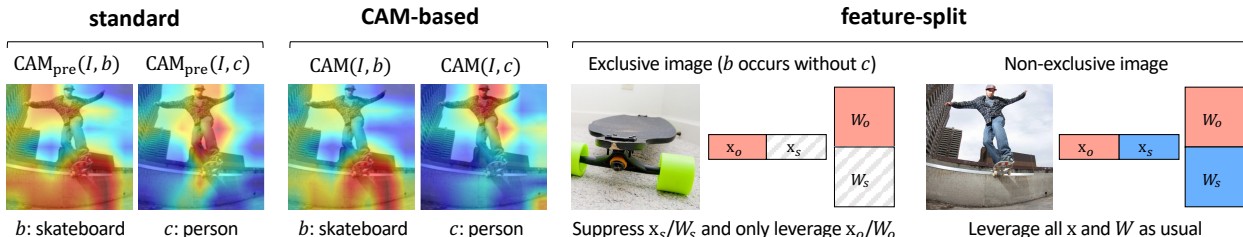

Figure 1: Overview of the proposed methods. The *CAM-based* methods enforces a minimal overlap between the $(b, c)$ CAMs, while preventing them from drifting too far from $\text{CAM}_{\text{pre}}$ (CAMs of the *standard* model). The *feature-split* method suppresses context for exclusive images by disabling backpropagation through $W_s$ and setting $x_s$ to a constant value; for non-exclusive images, it uses everything as usual.

### 3.1 The first proposed *CAM-based* method

The *CAM-based* method operates on the following premise: as $b$ almost always co-occurs with $c$, the network may learn to inadvertently rely on pixels corresponding to $c$ to predict $b$. The paper hypothesizes that one way to overcome this issue is to explicitly force the network to rely less on $c$'s pixel regions. This method uses class activation maps (CAMs) [12] as a proxy for object localization information. For an image $I$ and category $r$, $\text{CAM}(I, r)$ indicates the discriminative image regions used by a deep network to identify $r$. For each biased category pair $(b, c)$, a minimal overlap of their CAMs is enforced via the loss term:

$$L_O = \sum_{I \in \mathbb{I}_b \cap \mathbb{I}_c} \text{CAM}(I, b) \odot \text{CAM}(I, c), \tag{1}$$

where $\odot$ denotes element-wise multiplication and $\mathbb{I}_b \cap \mathbb{I}_c$ is a set of images where both $b$ and $c$ appear. To prevent a trivial solution where the CAMs of $b$ and $c$ drift apart from the actual pixel regions, the paper uses a regularization term to keep the category's CAMs close to $\text{CAM}_{\text{pre}}$, produced using a separate network trained offline:

$$L_R = \sum_{I \in \mathbb{I}_b \cap \mathbb{I}_c} |\text{CAM}_{\text{pre}}(I, b) - \text{CAM}(I, b)| + |\text{CAM}_{\text{pre}}(I, c) - \text{CAM}(I, c)|. \tag{2}$$

In our implementation, we separate a batch into two small batches during training, one with and one without co-occurrences. A sample is put into the *co-occurrence* batch if any of the 20 biased categories co-occurs with its context. For the co-occurrence batch, we compute CAM with the current model being trained and $\text{CAM}_{\text{pre}}$ with the trained

*standard* model, using the official CAM implementation: `https://github.com/zhoubolei/CAM`. We update the model parameters with the following loss, where $L_{\text{BCE}}$ is the binary cross entropy loss:

$$L_{\text{CAM}} = \lambda_1 L_O + \lambda_2 L_R + L_{\text{BCE}}. \tag{3}$$

For the *other* batch without any co-occurrences, we update the model parameters with $L_{\text{BCE}}$. With the hyperparameters reported in the paper, $\lambda_1 = 0.1$ and $\lambda_2 = 0.01$, we got underwhelming results and degenerate CAMs that drifted far from the actual pixel regions. Hence, we tried increasing the regularization weight $\lambda_2$ (0.01, 0.05, 0.1, 0.5, 1.0, 5.0) and achieved the best results with $\lambda_2 = 0.1$, which are reported in Table 2.

### 3.2 The second proposed *feature-split* method

By discouraging mutual spatial overlap, the *CAM-based* approach may not be able to leverage useful information from the pixel regions surrounding the context. Thus, the paper proposes a second method that splits the feature space into two subspaces to separately represent category and context, while posing no constraints on their spatial extents. Specifically, they propose using a dedicated feature subspace to learn examples of biased categories appearing without their context.

Given a deep neural network, let x denote the $D$-dimensional output of the final pooling layer just before the fully-connected (fc) layer. Let the weight matrix associated with fc layer be $W \in R^{D \times M}$, where $M$ denotes the number of categories given a multi-label dataset. The predicted scores inferred by a classifier (ignoring the bias term) are $\hat{y} = W^T x$. To separate the feature representations of a biased category from its context, the paper does a random row-wise split of $W$ into two disjoint subsets: $W_o$ and $W_s$ (dimension $\frac{D}{2} \times M$).[3] Consequently, x is split into $x_o$ and $x_s$, and $\hat{y} = W_o^T x_o + W_s^T x_s$. When a biased category occurs without its context, the paper disables backpropagation through $W_s$, forcing the network to learn only through $W_o$, and set $x_s$ to $\bar{x}_s$ (the average of $x_s$ over the last 10 mini-batches).

We implemented the *feature-split* method based on additional discussions with the original authors, to ensure that we replicated their method as closely as possible. For a single training batch, we first forwarded the entire batch through the model to obtain one set of scores $\hat{y}_{\text{non-exclusive}} = W_o^T x_o + W_s^T x_s$ and the corresponding features from the avgpool layer, which directly precedes the fc layer. We made a separate copy of these features and replaced $x_s$ with $\bar{x}_s$, then calculated a new set of output scores $\hat{y}_{\text{exclusive}} = W_o^T x_o + W_s^T \bar{x}_s$. Separate loss tensors for each of these outputs were computed, and elements corresponding to the exclusive and non-exclusive examples in the unmodified and modified loss tensors were zeroed out, respectively. The final loss tensor was obtained by adding these two together, and standard backpropagation was done using this final loss tensor. The gradients were calculated with respect to a weighed binary cross entropy loss:

$$L_{\text{WBCE}} = -\alpha \left[ t \log(\sigma(\hat{y})) + (1 - t) \log(1 - \sigma(\hat{y})) \right], \tag{4}$$

where $t$ is the ground-truth label, $\sigma$ is the sigmoid function, and $\alpha$ is the ratio between the number of training images in which a biased category occurs in the presence of its context and the number of images in which it occurs in the absence of its context. A higher value of $\alpha$ indicates more data skewness.[4]

### 3.3 Strong baselines

In addition to the *standard* model, the paper compares the proposed methods with several competitive *strong baselines*.

1. **Remove co-occur labels**: For each $b$, remove the $c$ label for images in which $b$ and $c$ co-occur.
2. **Remove co-occur images**: Remove training instances where any $b$ and $c$ co-occur. For COCO-Stuff, this process removes 29,332 images and leaves 53,451 images in the training set.
3. **Split-biased**: Split each $b$ into two classes: 1) $b \setminus c$ and 2) $b \cap c$. Unlike other "stage 2" models, this model is trained from scratch rather than on top of the *standard* baseline because it has 20 additional classes. We later confirmed with the authors that they did the same.
4. **Weighted loss**: For each $b$, apply 10 times higher weight to the loss for class $b$ when $b$ occurs exclusively.
5. **Negative penalty**: For each $(b, c)$, apply a large negative penalty to the loss for class $c$ when $b$ occurs exclusively. In our email communication, the authors said that the negative penalty means a 10 times higher weight to the loss.

---

[3]In an email, the authors noted that a random split is not critical; they obtained similar results with a random split and a middle split. We observed that a middle split of $W$ yields better results for COCO-Stuff and DeepFashion, but the opposite for AwA. As the gains from using a middle split for COCO-Stuff and DeepFashion were larger than the losses for AwA, we chose to use a middle split.

[4]In practice, the paper ensures $\alpha$ is at least $\alpha_{\min}$, which they set to 3 for COCO-Stuff and AwA and 5 for DeepFashion. However, we found that most of the paper's biased category pairs have $\alpha$ smaller than $\alpha_{\min}$. Out of 20 pairs, 13 pairs for COCO-Stuff, 20 pairs for DeepFashion, and 19 pairs for AwA had $\alpha$ smaller than $\alpha_{\min}$. We also tried using higher values of $\alpha_{\min}$ but didn't gain meaningful improvements, so we report results with the original authors' $\alpha_{\min}$.

Table 2: Performance of different methods on COCO-Stuff, DeepFashion, AwA, and UnRel on "exclusive" and "co-occur" distributions with best results in bold. We compare our results to the paper's results, specifically its Table 2, 3, 4, 5, 8, 9. Per-category results can be found in Appendix H.

| Method | COCO-Stuff (mAP) | | | | DeepFashion (top-3 recall) | | | | AwA (mAP) | | | | UnRel (mAP) | |
| --- | --- | --- | --- | --- | --- | --- | --- | --- | --- | --- | --- | --- | --- | --- |
| | Exclusive | | Co-occur | | Exclusive | | Co-occur | | Exclusive | | Co-occur | | 3 categories | |
| | Paper | Ours | Paper | Ours | Paper | Ours | Paper | Ours | Paper | Ours | Paper | Ours | Paper | Ours |
| *standard*[5] | 24.5 | 23.9 | **66.2** | 65.0 | 4.9 | 7.0 | 17.8 | 22.8 | 19.4 | 19.5 | 72.2 | 69.0 | 42.0 | 43.0 |
| *remove labels* | 25.2 | 24.5 | 65.9 | 64.6 | 6.0 | 7.5 | **20.4** | 24.4 | 19.1 | 18.9 | 62.9 | 63.2 | - | 42.7 |
| *remove images* | 28.4 | **29.0** | 28.7 | 59.6 | 4.2 | 5.6 | 5.4 | 13.0 | **22.7** | **21.7** | 58.3 | 65.2 | - | 48.6 |
| *split-biased* | 19.1 | 25.4 | 64.3 | 64.7 | 3.5 | 4.9 | 14.3 | 11.1 | 19.7 | 18.2 | 66.8 | 64.2 | - | 29.2 |
| *weighted* | **30.4** | 28.5 | 60.8 | 60.0 | - | **29.5** | - | **43.6** | - | 20.0 | - | 67.7 | - | 44.4 |
| *negative penalty* | 23.8 | 23.9 | 66.1 | 64.7 | 5.5 | 7.8 | 18.9 | 23.8 | 19.2 | 19.6 | 68.4 | 69.0 | - | 42.5 |
| *class-balancing* | 25.0 | 24.6 | 66.1 | 64.7 | 5.2 | 8.0 | 19.4 | 24.8 | 20.4 | 19.9 | 68.4 | 68.2 | - | 42.3 |
| *attribute decorr.* | - | - | - | - | - | - | - | - | 18.4 | 20.6 | 70.2 | **69.8** | - | - |
| *CAM-based* | 26.4 | 26.9 | 64.9 | 64.2 | - | - | - | - | - | - | - | - | 45.3 | 46.8 |
| *feature-split* | 28.8 | 28.1 | 66.0 | 64.8 | **9.2** | 12.2 | 20.1 | 27.1 | 20.8 | 19.2 | **72.8** | 68.6 | **52.1** | **49.9** |

Figure 2: A visual comparison of the results on COCO-Stuff from Table 2. The blue and red lines mark the paper's and our *standard* mAPs. Similar plots for DeepFashion and AwA can be found in Appendix F.

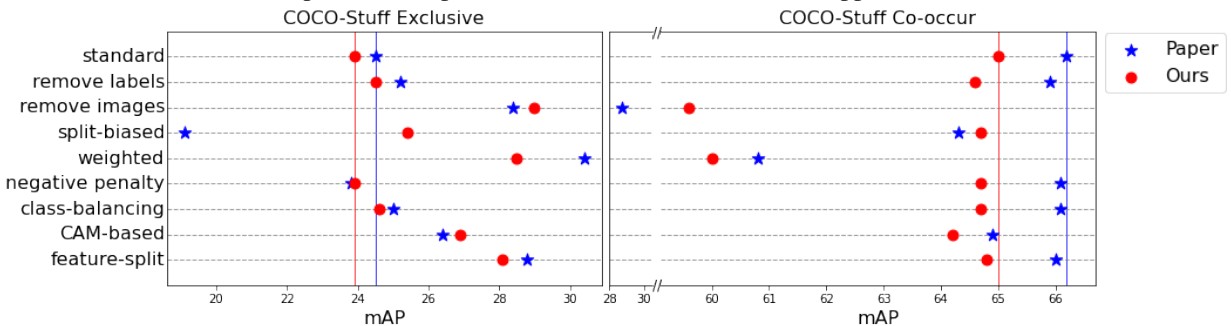

6. **Class-balancing loss** [2]: For each $b$, put the images in three groups: exclusive, co-occurring, and other. The weight for each group is $(1 - \beta)/(1 - \beta^n)$ where $n$ is the number of images for each group and $\beta$ is a hyperparameter. The authors said they set $\beta = 0.99$ in our email communication.

7. **Attribute decorrelation** [5]: Use the proposed method, but replace the hand-crafted features used in [5] with deep network features (i.e., conv5 features of a trained "stage 1" ResNet-50).

### 3.4 Training details and computational requirements

We trained all "stage 2" models on top of the *standard* model for 20 epochs using a learning rate of 0.01, a batch size of 200, and SGD with 0.9 momentum. The exceptions are *split-biased* which is not trained on top of the *standard* model and is thus trained for an additional 20 epochs to ensure a fair comparison; and *CAM-based* which uses a batch size of 100 due to memory limits. All models were trained on a single RTX 3090 GPU and evaluated on the last epoch. On COCO-Stuff, the single-epoch training time was around 12.9 minutes for *standard*, *remove labels*, *split-biased*, *weighted*, *negative penalty*, and *class-balancing*. It took 8.4 minutes to train *remove images*, and 17.3 minutes and 13.3 minutes to train *CAM-based* and *feature-split*, respectively. Thus, we reach a different conclusion from the paper's claim that the "overall training time of both proposed methods is very close to that of a standard classifier." We suspect that this difference is due to the difference in implementation. Overall, the total training time for each method range from 35-43 hours on COCO-Stuff, 22-29 hours on DeepFashion, and 7-8 hours on AwA. For inference, the paper reports that a single forward pass of an image takes 0.2ms on a single Titan X GPU for the *standard*, *CAM-based*, and *feature-split* methods. We confirmed that it takes the same amount of time for the three methods. See Appendix E for detailed training and inference times.

### 3.5 Results

In Table 2, we compare the performance of the ten methods, evaluated with the paper's biased category pairs for consistency. Additiional figures and per-category results can be found in Appendix F and H.

---

[5]To ensure a fair comparison with the "stage 2" models, we tried training the *standard* model for an additional 20 epochs but did not see improvements; hence, we report the *standard* results from Table 1.

**COCO-Stuff:** Since our *standard* model underperforms the paper's by 0.6-3.1% (Section 2), we focus on the relative ordering between the different methods visualized in Figure 2. In the paper, all but *split-biased* and *negative penalty* improve upon the *standard* baseline's "exclusive" mAP; whereas in our experiments, only *negative penalty* fails to improve on *standard*'s "exclusive" mAP. Different from the paper, *remove images* has the highest "exclusive" mAP in our experiments, followed by *weighted* and the paper's proposed methods, *feature-split* and *CAM-based*. For *feature-split*, we observed a similar tradeoff between "exclusive" and "co-occur" mAPs compared to the paper. All methods have similar performance of 55.0–55.7 mAP when evaluated on the full test set for all 171 categories.

**DeepFashion:** Consistent with the paper, all methods except *remove images* and *split-biased* improve upon *standard*'s "exclusive" top-3 recall. We found the *weighted* method performs the best out of all the methods with +22.5% for "exclusive" and +20.8% for "co-occur." However, it has a relatively low top-3 recall when evaluated on the full test set for all 250 categories: 23.3 compared to other methods' top-3 recall in the range of 23.8–24.3.

**Animals with Attributes:** Unlike the result reported in the paper, our reproduced *feature-split* model had a -0.3% drop in "exclusive" mAP and a -0.4% drop in "co-occur" mAP compared to the *standard* model. In Section 3.6, when we experimented with different subspace sizes, we observed that the *feature-split* model trained with $x_o$ of size 1,792 improves upon the *standard* model on both test distributions. Among all the methods, *remove images* improves the "exclusive" mAP the most; however, this method also suffers from a noticeable decrease in "co-occur" performance. When evaluated on the full test set for all 85 categories, most methods have similar mAP in the range of 72.5–73.0, except for *remove labels* that has 70.6 mAP and *remove images* that has 69.7 mAP.

**UnRel:** The paper includes a cross-dataset experiment where the models trained on COCO-Stuff are applied without any fine-tuning on UnRel, a dataset that contains images of objects outside of their typical context. The paper evaluates the models only on the 3 categories of UnRel that overlap with the 20 most biased categories of COCO-Stuff, which we determined to be skateboard, car, and bus. While the paper does not report results from the *remove images* baseline, for us it had the highest mAP of the 3 categories, followed by the *feature-split* and *CAM-based* methods.

### 3.6  Additional analyses

**Cosine similarity between $W_o$ and $W_s$:** The paper computes the cosine similarity between $W_o$ and $W_s$ to investigate if they capture distinct sets of information. It reports that the proposed methods yield a lower similarity score compared to the *standard* model, and concludes that the biased class $b$ is less dependent on $c$ for prediction in their methods. To reproduce their results, for *feature-split*, we calculated the cosine similarity between $W_o[:, b]$ and $W_s[:, b]$ (dimensions $\frac{D}{2}$) for each $b$ of the 20 $(b, c)$ pairs and reported their average. On the other hand, $W_o$ and $W_s$ are not specified for *standard* and *CAM-based*. Hence, we randomly split $W$ in half and defined one as $W_o$ and the other as $W_s$.

In Table 3, we compare our reproduced results with the paper's results. Consistent with the paper's conclusion, we find that the proposed methods have weights with similar or lower cosine similarity. On the interpretation of the results, we agree that *feature-split*'s low cosine similarity suggests that the corresponding feature subspaces $x_o$ and $x_s$ capture different information, as intended by the method. However, we don't understand why the cosine similarity of *CAM-based* would be lower than *standard*, as there is nothing in *CAM-based* that encourages the feature subspaces to be distinct.

| Method | COCO-Stuff | | DeepFashion | | AwA | |
|---|---|---|---|---|---|---|
| | Paper | Ours | Paper | Ours | Paper | Ours |
| *standard* | 0.21 | 0.08 | - | 0.12 | - | 0.02 |
| *CAM-based* | 0.19 | 0.07 | - | - | - | - |
| *feature-split* | 0.17 | 0.04 | - | 0.05 | - | 0.02 |

Table 3: Cosine similarity between $W_o$ and $W_s$ for the 20 most biased categories. We compare our reproduced results to those in the paper's Table 7. The paper does not report results for the DeepFashion and AwA datasets.

**Qualitative analysis:** Following Section 5.1.2 of the original paper, we used CAMs to visually analyze the proposed methods. In general, our observations are in line with those of the original paper. For example, in Figure 3, we see that *CAM-based* tends to only focus on the right pixel regions (e.g., skateboard, microwave) compared to *standard*, while *feature-split* also makes use of context (e.g., person, oven). More analyses are available in Appendix G.

## 4  Our additional experiments

To better understand the proposed *CAM-based* and *feature-split* methods, we conducted several ablation studies (Table 4).

Figure 3: Biased category CAMs for (skateboard, person) and (microwave, oven) pairs.

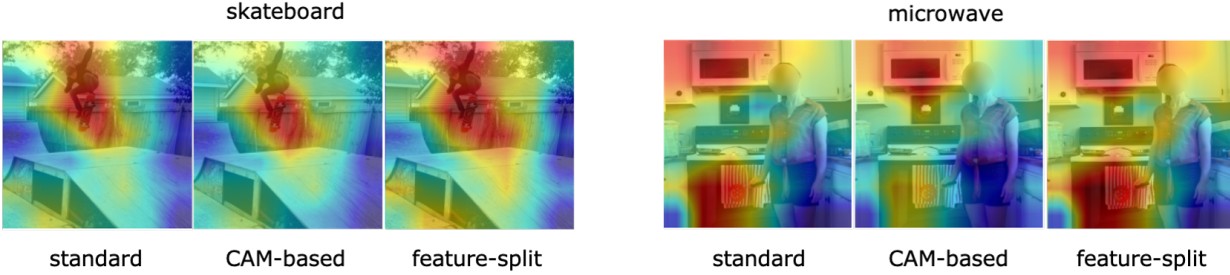

**What is the effect of the regularization term in the *CAM-based* method?** As mentioned in Section 3.1, we tried varying the weight for the regularization term $L_R$ ($\lambda_2$) in the *CAM-based* method that prevents the CAMs of the biased category pairs from drifting apart from the pixel regions of $CAM_{pre}$. We observed that weak regularization allows for highly localized, degenerate CAMs that don't resemble $CAM_{pre}$, while overly strong regularization makes the method less effective. We were able to strike an ideal balance with $\lambda_2 = 0.1$, higher than the paper's $\lambda_2 = 0.01$.

**What is the effect of the weighted loss in the *feature-split* method?** To understand the effect of the weighted loss in the *feature-split* method, we tried training a *feature-split* model without it and a baseline model with the *feature-split* weighted loss. Both variations have lower "exclusive" mAPs, suggesting that both the feature-splitting framework and the weighted loss are important components of the method. We highlight that the *feature-split* model trained without the weighted loss is worse than the *standard* model, suggesting that the weighted loss is central for the *feature-split* method to achieve good performance. However, we also observed that the *feature-split* method's weighted loss by itself is not sufficient for improving the performance of the *standard* model on the "exclusive" distribution.

**Does the size of the *feature-split* subspace matter?** In the *feature-split* method, the original paper allocates half of the 2,048 feature dimensions in the fc layer for learning exclusive image examples. We explored whether a smaller or larger $x_o$ subspace may strike a better balance and improve both "exclusive" and "co-occur" performance, as the number of exclusive images is only a small fraction of the entire training data. For COCO-Stuff, the performance peaks on "exclusive" and dips on "co-occur" at the 1,024 dimension split. For DeepFashion, performance on both distributions peak at the 1,024 dimension split. For AwA, however, the performance on both distributions improves as the subspace size increases. Lastly for UnRel, the model trained on COCO-stuff with a $x_o$ of size 768 performs best. Overall, we did not find a clear trend between *feature-split* performance and subspace size.

Table 4: (Top) Ablation studies of *CAM-based* and *feature-split* on COCO-Stuff. (Bottom) Additional *feature-split* results with varying $x_o$ subspace sizes. Best results are in bold.

| Method | Exclusive | Co-occur | All | Non-biased |
|---|---|---|---|---|
| *standard* | 23.9 | **65.0** | **55.7** | **72.3** |
| *CAM-based* with $\lambda_2 = 0$ (no regularization) | 24.4 | 64.6 | 55.5 | 72.0 |
| *CAM-based* with $\lambda_2 = 0.01$ (paper params) | 24.6 | 64.6 | 55.5 | 72.0 |
| *CAM-based* with $\lambda_2 = 0.1$ (tuned params) | **26.9** | 64.2 | 55.5 | 72.2 |
| *feature-split* | **28.1** | 64.8 | 55.6 | 72.1 |
| *feature-split* without weighted loss | 23.6 | 65.4 | 55.6 | 72.1 |
| baseline with *feature-split* weighted loss | 24.0 | 64.8 | 55.5 | 72.1 |

| $x_o$ size | COCO-Stuff (mAP) | | DeepFashion (top-3 recall) | | AwA (mAP) | | UnRel (mAP) |
|---|---|---|---|---|---|---|---|
| | Exclusive | Co-occur | Exclusive | Co-occur | Exclusive | Co-occur | 3 categories |
| 256 | 23.8 | 65.9 | 3.2 | 12.5 | 18.5 | 69.7 | 47.8 |
| 512 | 26.7 | 65.9 | 3.4 | 12.9 | 18.7 | 69.0 | 50.1 |
| 768 | 27.2 | 65.8 | 4.6 | 14.3 | 19.1 | 69.2 | **50.4** |
| 1,024 | **28.1** | 64.8 | **12.2** | **27.1** | 19.2 | 68.6 | 49.9 |
| 1,280 | 24.8 | 66.0 | 4.0 | 13.7 | 19.3 | 69.9 | 45.8 |
| 1,536 | 23.0 | 66.1 | 2.7 | 13.9 | 19.5 | 70.3 | 44.2 |
| 1,792 | 21.7 | **66.2** | 2.3 | 14.1 | **19.7** | **70.8** | 40.6 |

# 5    Discussion

We found that the proposed *CAM-based* and *feature-split* methods help mitigate contextual bias, although we could not completely replicate the quantitative results in the original paper even after completing an extensive hyperparameter search. As an effort to check our conclusions, we tried several different approaches in how we choose our best models, train the baselines, and performed evaluation. We also conducted additional analyses of the proposed methods to check our implementations and train them to achieve their best possible performance. In all cases, decreasing contextual bias frequently came with the cost of decreasing performance on non-biased categories. Ultimately, we believe deciding what method is best depends on the trade-offs a user is willing to make in a given scenario, and the original paper's proposed methods seem to strike a good balance for the tested datasets.

**Recommendations for reproducibility:** Overall, the paper was clearly written and it was easy to follow the explanation and reasoning of the experiments. Still, we ran into several obstacles while re-implementing the entire pipeline from scratch. Our biggest concern was making sure that most, if not all, training/evaluation details were true to the experiments in the paper. We are extremely grateful to the original authors who gave swift responses to our questions. Nevertheless, it would have been easier to reproduce the results with code or a README file listing design decisions. Given the limited information, it took us over a month to lock in various details on data processing, hyperparameter optimization, and training the *standard* model, before we could move onto reproducing the "stage 2" methods. Moreover, each method had its intricacies and we inevitably ran into ambiguities along the way. For example, the *attribute decorrelation* method took considerable time to reproduce because no hyperparameters or code were given in the paper or the original work [5]. We hope our report and published code help future use of the paper.

**Recommendations for reproducing papers:** In closing, we would like to share a few things that we found helpful as suggestions for future reproducibility efforts. First, writing the mandatory reproducibility plan (provided in Section I of the appendix) at the beginning of the challenge was helpful, as it forced us to define concrete steps for reproducing the experiments. We suggest putting together a similar plan because the order in which materials are presented in the paper can be different from the order in which experiments should be run. Additionally, we recommend communicating early with the original authors to determine undisclosed parameters and pin down the experimental setup. Lastly, for reproducing training processes in particular, we suggest checking how training is progressing in as many different ways as possible. In our process, this involved looking at the progression of CAMs and examining training curves for individual loss function terms, both of which helped us pinpoint our issues.

## Acknowledgements

This work is supported by the National Science Foundation under Grant No. 1763642 and the Princeton First Year Fellowship to SK. We thank the authors of the original paper, especially the lead author Krishna Kumar Singh, who gave detailed and swift responses to our questions. We also thank Angelina Wang, Felix Yu, Vikram Ramaswamy, Vivien Nguyen, Zeyu Wang, and Zhiwei Deng for helpful comments and suggestions.

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

# Appendix

We dedicate the appendix to providing more details on certain parts of the main paper.

- In Section A, we describe how we obtained and processed the four datasets.
- In Section B, we provide additional details on biased categories identification.
- In Section C, we describe our hyperparameter search.
- In Section D, we discuss different model selection methods we tried while reproducing the *standard* baseline.
- In Section E, we provide more details on computational requirements.
- In Section F, we provide visualizations of DeepFashion and AwA results.
- In Section G, we provide additional qualitative analyses with CAMs.
- In Section H, we provide per-category results for COCO-Stuff, DeepFashion, Animals with Attributes, and UnRel.
- In Section I, we provide the reproducibility plan we wrote at the start of the project.

## A  Datasets

In this section, we describe how we obtained and processed the four datasets used in the paper. COCO-Stuff [1] and UnRel [8] are used for the object classification task, and DeepFashion [7] and Animals with Attributes [10] are used for the attribute classification task. COCO-Stuff is the main dataset used for discussion of quantitative and qualitative results. UnRel is used for cross-dataset experiments, i.e. testing models trained on COCO-Stuff on UnRel without fine-tuning.

### A.1  COCO-Stuff

We downloaded COCO-Stuff [1] from the official homepage: `https://github.com/nightrome/cocostuff`. COCO-Stuff includes all 164K images from COCO-2017 (train 118K, val 5K, test-dev 20K, test-challenge 20K), but only the training and validation set annotations are publicly available. It covers 172 classes: 80 thing classes, 91 stuff classes and 1 class designated 'unlabeled.'

COCO-Stuff (COCO-2017 with "stuff" annotations added) contains the same images as COCO-2014 [6] but has different train-val-test splits. The original paper follows the data split of COCO-2014 and uses 82,783 images for training and 40,504 images for evaluation. The image numbers are consistent between COCO-2014 and COCO-2017, so we were able to map the "stuff" annotations from COCO-Stuff to the COCO-2014 images with "thing" annotations. Excluding the 'unlabeled' category, we have in total 171 categories.

In Table A1, we report the co-occurrence, exclusive, and other counts for the paper's 20 biased category pairs. The co-occurrence count is the number of images where $b$ and $c$ co-occur; the exclusive count is the number of images where $b$ occurs without $c$; the other count is the number of remaining images where $b$ doesn't occur.

During our data processing, we found a small typo in the original paper. Section 3 of the paper says "COCO-Stuff has 2,209 images where 'ski' co-occurs with 'person,' but only has 29 images where 'ski' occurs without 'person.'" On the other hand, we found 2,180 co-occurring and 29 exclusive images in the training set. We verified with the authors that our data processing was correct. Merging COCO-2014 and COCO-Stuff annotations is a nontrivial step in the pipeline. We hope our published code and the Table A1 help future use.

### A.2  DeepFashion

We downloaded DeepFashion [7] by following in the instructions on the official homepage: `http://mmlab.ie.cuhk.edu.hk/projects/DeepFashion.html`. The dataset consists of 5 benchmarks, out of which we use the Category and Attribute Prediction Benchmark. This benchmark consists of 209,222 training images, 40,000 validation images, and 40,000 test images with 1,000 attribute classes in total. Per the procedure specified by the authors, we only use the 250 most commonly appearing attributes. In Table A2, we report the co-occur, exclusive and other counts for the paper's 20 biased category pairs. It should be noted that the DeepFashion dataset was updated with additional "fine-grained attribute annotations" in May 2020.

### A.3  Animals with Attributes

Animals with Attributes (AwA) [10] is suspended and the images are no longer available because of copyright restrictions, according to the official homepage: `https://cvml.ist.ac.at/AwA/`. Hence we downloaded Animals with Attributes 2 (AwA2), which is described as a "drop-in replacement" to AwA as it has the same class structure and almost the same characteristics, from the AwA2 official homepage: `https://cvml.ist.ac.at/AwA2/`. We confirmed with

the authors that they used AwA2 as well. AwA2 consists of 30,337 training images with 40 animal classes and 6,985 test images with 10 other animal classes, with pre-extracted feature representations for each image. The classes are aligned with Osherson's classical class/attribute matrix, thereby providing 85 numeric attribute values for each class. The images were collected from public sources, such as Flickr, in 2016.

In Table A3, we report the co-occurrence, exclusive, and other counts for the paper's 20 biased category pairs. Following the description in the paper, we trained all models on the training set (40 classes) and evaluate on the test set (10 classes). For biased categories identification, following the paper description, we used the test set to determine the biased categories as these two sets contain different attribute distributions.

## A.4 UnRel

We downloaded UnRel [8] from the official homepage: `https://github.com/jpeyre/unrel`. This dataset contains 1,071 images of objects out of their typical context and serves as a stress test for the models trained on COCO-Stuff. According to the paper, there are only three categories in UnRel that are shared with the 20 biased categories found in COCO-Stuff. We determined these categories to be "skateboard," "car" and "bus." Only these three categories were used in the evaluation.

## B  Biased categories identification

In this section, we provide additional details on the biased categories identification process discussed in Section 2.2 of the main paper.

For each dataset, the paper identifies the top-20 $(b, c)$ pairs of biased categories, where $b$ is the category suffering from contextual bias and $c$ is the associated context category. For a given category $z$, let $\mathbb{I}_b \cap \mathbb{I}_z$ and $\mathbb{I}_b \setminus \mathbb{I}_z$ denote sets of images where $b$ occurs with and without $z$ respectively. Let $\hat{p}(I, b)$ denote the prediction probability of an image $I$ for a category $b$ obtained from a trained multi-class classifier. The *bias* between two categories $b$ and $z$ is defined as follows:

$$\text{bias}(b, z) = \frac{\frac{1}{|\mathbb{I}_b \cap \mathbb{I}_z|} \sum_{I \in \mathbb{I}_b \cap \mathbb{I}_z} \hat{p}(I, b)}{\frac{1}{|\mathbb{I}_b \setminus \mathbb{I}_z|} \sum_{I \in \mathbb{I}_b \setminus \mathbb{I}_z} \hat{p}(I, b)}, \tag{5}$$

which is the ratio of average prediction probabilities of $b$ when it occurs with and without $z$. The category $c$ that most biases $b$ is determined as $c = \arg\max_z \text{bias}(b, z)$, with a condition that they co-occur frequently. Specifically, the paper defines that $b$ must co-occur at least 20% of the time with $c$ for COCO-Stuff and AwA, and 10% for DeepFashion. In

---

[6]We found this vague as there are two ceiling categories in COCO-Stuff: ceiling-other and ceiling-tile. We interpreted it as ceiling-other as ceiling-tile doesn't frequently co-occur with toaster.

| Biased category pairs | | Bias | | Training (82,783) | | Test (40,504) | | Biased category pairs (Ours) | | |
|---|---|---|---|---|---|---|---|---|---|---|
| Biased ($b$) | Context ($c$) | Paper | Ours | Co-occur | Exclusive | Co-occur | Exclusive | Biased ($b$) | Context ($c$) | Bias |
| cup | dining table | 1.76 | 1.85 | 3,186 | 3,140 | 1,449 | 1,514 | car | road | 1.73 |
| wine glass | person | 1.80 | 1.59 | 1,151 | 583 | 548 | 304 | potted plant | furniture-other | 1.75 |
| handbag | person | 1.81 | 2.25 | 4,380 | 411 | 2,035 | 209 | spoon | bowl | 1.75 |
| apple | fruit | 1.91 | 2.12 | 477 | 627 | 208 | 244 | fork | dining table | 1.78 |
| car | road | 1.94 | 1.73 | 5,794 | 2,806 | 2,842 | 1,331 | bus | road | 1.79 |
| bus | road | 1.94 | 1.79 | 2,283 | 507 | 1,090 | 259 | cup | dining table | 1.85 |
| potted plant | vase | 1.99 | 1.73 | 930 | 2,152 | 482 | 1,058 | mouse | keyboard | 1.87 |
| spoon | bowl | 2.04 | 1.75 | 1,314 | 954 | 638 | 449 | remote | person | 1.89 |
| microwave | oven | 2.08 | 1.59 | 632 | 450 | 291 | 217 | wine glass | dining table | 1.94 |
| keyboard | mouse | 2.25 | 2.11 | 860 | 601 | 467 | 278 | clock | building-other | 1.97 |
| skis | person | 2.28 | 2.21 | 2,180 | 29 | 984 | 9 | keyboard | mouse | 2.11 |
| clock | building | 2.39 | 1.97 | 1,410 | 1,691 | 835 | 840 | apple | fruit | 2.12 |
| sports ball | person | 2.45 | 3.61 | 2,607 | 105 | 1,269 | 55 | skis | snow | 2.22 |
| remote | person | 2.45 | 1.89 | 1,469 | 666 | 656 | 357 | handbag | person | 2.25 |
| snowboard | person | 2.86 | 2.40 | 1,146 | 22 | 522 | 11 | snowboard | person | 2.40 |
| toaster | ceiling[6] | 3.70 | 1.98 | 60 | 91 | 30 | 44 | skateboard | person | 3.41 |
| hair drier | towel | 4.00 | 3.49 | 54 | 74 | 28 | 41 | sports ball | person | 3.61 |
| tennis racket | person | 4.15 | 1.26 | 2,336 | 24 | 1,180 | 10 | hair drier | sink | 6.11 |
| skateboard | person | 7.36 | 3.41 | 2,473 | 38 | 1,068 | 24 | toaster | oven | 8.56 |
| baseball glove | person | 339.15 | 31.32 | 1,834 | 19 | 820 | 9 | baseball glove | person | 31.32 |

Table A1: (Left) The paper's 20 most biased category pairs for **COCO-Stuff** and their bias values, both what's reported in the paper and what we've calculated with our trained model. (Middle) The number of co-occuring and exclusive images for each pair. (Right) The 20 most biased categories we've identified with our trained model.

short, a given category $b$ is most biased by $c$ if (1) $b$ co-occurs frequently with $c$ and (2) the prediction probability of $b$ drop significantly in the *absence* of $c$.

While this method can be applied to any number of biased category pairs, the paper says using $K = 20$ sufficiently captures biased categories in all datasets used the paper. We report the 20 most biased category pairs we've identified and compare them to those identified by the paper in Tables A1 (COCO-Stuff), A2 (DeepFashion), A3 (AwA). We discuss the results for each dataset in more detail below.

**COCO-Stuff:** Overall, the bias values of the paper's biased category pairs calculated with our model are similar to the paper's values. Furthermore, most of our biased category pairs match with the paper's pairs. 18 of the 20 biased categories overlap, although their context categories sometimes differ.

**DeepFashion:** After manual cleaning per suggestion of the authors, 10 of our biased category pairs match with the paper's. Still, the bias values of the paper's pairs calculated with our trained model are overall similar to the paper's values. It is worth noting that there are fewer co-occurring and exclusive images for each of the biased category pairs, compared to COCO-Stuff.

**Animals with Attributes:** Almost all of our biased categories match with those in the paper. We did observe in the process of determining the biased categories that for each $b$, there were multiple categories $c$ which had an equally biased effect on $b$. That is, the bias value bias($b, c$) was equal over each of these $c$'s. We suspect that this is because the images in AwA are labeled by animal class rather than per image, so many images share the same exact labels. Moreover, we observed that for many image examples, the baseline model's highest prediction scores differ by less than 0.001 or even 0.0001. The combination of these two events may result in extremely similar bias scores. Since there were multiple $c$'s for each $b$, we listed the category which matched the paper's findings whenever possible. In total, 18 of our biased categories overlapped with those in the paper.

# C Hyperparameter search

In this section, we describe how we conducted our hyperparameter search. The paper does not describe the hyperparameter search process, so we followed standard practice and tuned the hyperparameters on the validation set. While DeepFashion has training, validation and test sets, COCO-Stuff and AwA don't have validation sets, so we created a random 80-20 split of the original training set and used the 80 split as the training set and the 20 split as the validation set. We later confirmed with the authors that this is how they did their hyperparameter search.

| Biased category pairs | | Bias | | Training (209,222) | | Test (40,000) | | Biased category pairs (Ours) | | |
|---|---|---|---|---|---|---|---|---|---|---|
| Biased ($b$) | Context ($c$) | Paper | Ours | Co-occur | Exclusive | Co-occur | Exclusive | Biased ($b$) | Context ($c$) | Bias |
| bell | lace | 3.15 | 2.74 | 167 | 549 | 32 | 92 | boyfriend | distressed | 3.35 |
| cut | bodycon | 3.30 | 3.46 | 313 | 2612 | 58 | 488 | gauze | embroidered | 3.35 |
| animal | print | 3.31 | 2.29 | 592 | 234 | 106 | 52 | la | muscle | 3.35 |
| flare | fit | 3.31 | 2.56 | 2,960 | 527 | 561 | 103 | diamond | print | 3.40 |
| embroidery | crochet | 3.44 | 3.04 | 237 | 1,021 | 42 | 221 | york | city | 3.43 |
| suede | fringe | 3.48 | 2.75 | 104 | 478 | 23 | 92 | retro | chiffon | 3.43 |
| jacquard | flare | 3.68 | 4.02 | 71 | 538 | 11 | 107 | cut | bodycon | 3.46 |
| trapeze | striped | 3.70 | 2.85 | 51 | 531 | 14 | 127 | fitted | sleeve | 3.58 |
| neckline | sweetheart | 3.98 | 3.16 | 161 | 818 | 25 | 156 | light | wash | 3.59 |
| retro | chiffon | 4.08 | 3.43 | 119 | 1,135 | 26 | 224 | sequin | mini | 3.63 |
| sweet | crochet | 4.32 | 6.55 | 180 | 1,122 | 29 | 190 | cuffed | denim | 3.70 |
| batwing | loose | 4.36 | 3.89 | 181 | 518 | 40 | 100 | lady | chiffon | 3.71 |
| tassel | chiffon | 4.48 | 3.15 | 71 | 651 | 8 | 131 | jacquard | fit | 4.02 |
| boyfriend | distressed | 4.50 | 3.35 | 276 | 1,172 | 63 | 215 | bell | sleeve | 4.23 |
| light | skinny | 4.53 | 3.31 | 216 | 1,621 | 47 | 298 | ankle | skinny | 4.42 |
| ankle | skinny | 4.56 | 4.42 | 340 | 462 | 68 | 96 | tiered | crochet | 4.45 |
| french | terry | 5.09 | 7.64 | 975 | 646 | 178 | 121 | studded | denim | 4.98 |
| dark | wash | 5.13 | 5.66 | 343 | 1,011 | 69 | 191 | dark | wash | 5.66 |
| medium | wash | 7.45 | 6.78 | 227 | 653 | 35 | 153 | sweet | crochet | 6.55 |
| studded | denim | 7.80 | 4.98 | 139 | 466 | 25 | 95 | medium | wash | 6.78 |

Table A2: (Left) The paper's 20 most biased category pairs for **DeepFashion** and their bias values, both what's reported in the paper and what we've calculated with our trained model. (Middle) The number of co-occuring and exclusive images for each pair. (Right) The 20 most biased categories we've identified with our trained model.

Table A3: (Left) The paper's 20 most biased category pairs for **AwA** and their bias values, both what's reported in the paper and what we've calculated with our trained model. (Middle) The number of co-occuring and exclusive images for each pair. (Right) The 20 most biased categories we've identified with our trained model.

| Biased category pairs | | Bias | | Training (30,337) | | Test (6,985) | | Biased category pairs (Ours) | | |
|---|---|---|---|---|---|---|---|---|---|---|
| Biased (b) | Context (c) | Paper | Ours | Co-occur | Exclusive | Co-occur | Exclusive | Biased (b) | Context (c) | Bias |
| white | ground | 3.67 | 4.08 | 12,952 | 1,237 | 3,156 | 988 | forager | nestspot | 4.04 |
| longleg | domestic | 3.71 | 6.55 | 3,727 | 7,667 | 728 | 720 | white | ground | 4.08 |
| forager | nestspot | 4.02 | 4.04 | 7,740 | 7,214 | 3,144 | 713 | hairless | swims | 4.29 |
| lean | stalker | 4.46 | 3.91 | 5,312 | 11,592 | 720 | 1,038 | muscle | black | 4.63 |
| fish | timid | 5.14 | 6.30 | 2,786 | 2,675 | 4,002 | 1,232 | insects | gray | 4.97 |
| hunter | big | 5.34 | 8.99 | 6,557 | 3,207 | 1,708 | 310 | fish | timid | 6.30 |
| plains | stalker | 5.40 | 1.81 | 3,793 | 12,865 | 720 | 310 | longleg | domestic | 6.55 |
| nocturnal | white | 5.84 | 6.97 | 3,118 | 2,464 | 822 | 720 | nocturnal | white | 6.97 |
| nestspot | meatteeth | 5.92 | 8.14 | 4,788 | 5,180 | 2,270 | 874 | nestspot | meatteeth | 8.14 |
| jungle | muscle | 6.26 | 9.15 | 4,480 | 696 | 2,132 | 874 | hunter | big | 8.99 |
| muscle | black | 6.39 | 4.63 | 10,656 | 8,960 | 2,157 | 684 | jungle | muscle | 9.15 |
| meat | fish | 7.12 | 10.17 | 3,175 | 7,819 | 1,979 | 310 | meat | fish | 10.17 |
| mountains | paws | 9.24 | 14.74 | 3,090 | 4,897 | 1,232 | 728 | domestic | inactive | 11.02 |
| tree | tail | 10.98 | 11.48 | 2,121 | 1,255 | 1,960 | 874 | tree | tail | 11.48 |
| domestic | inactive | 11.77 | 11.02 | 5,853 | 5,953 | 3,322 | 728 | spots | longleg | 12.50 |
| spots | longleg | 20.15 | 12.50 | 3,095 | 2,433 | 720 | 3,087 | mountains | paws | 14.74 |
| bush | meat | 29.47 | 31.26 | 1,896 | 5,922 | 6,265 | 1,602 | bush | meat | 31.26 |
| buckteeth | smelly | 34.01 | 51.25 | 3,701 | 3,339 | 310 | 874 | buckteeth | smelly | 51.25 |
| slow | strong | 76.59 | 125.19 | 8,710 | 1,708 | 3,968 | 747 | slow | strong | 125.19 |
| blue | coastal | 319.98 | 1,393.25 | 946 | 174 | 709 | 747 | blue | coastal | 1,393.25 |

**Search for the *standard* model:** For COCO-Stuff, we tried varying the learning rate (0.1, 0.05, 0.01), weight decay (0, 1e-5, 1e-4, 1e-3), and the epoch after which learning rate is dropped (20, 40, 60). We found that the paper's hyperparameters (0.1 learning rate dropped to 0.01 after epoch 60 with no weight decay) produced the best results. For DeepFashion, we varied the learning rate (0.1, 0.05, 0.01, 0.005, 0.001, 0.0001), weight decay (0, 1e-6, 1e-5, 1e-4), and the epoch after which the learning rate dropped (20, 30). We obtained the best results using a constant learning rate of 0.1 and weight decay of 1e-6. For AwA, we tried learning rates of 0.1 and 0.01, with various training schedules such as dropping from 0.1 to 0.001, dropping from 0.01 to 0.001, and keeping a constant learning rate of 0.01 throughout. We also tried varying weight decay (0, 1e-2, 1e-3, 1e-4, 1e-5), but the paper's hyperparameters (0.1 learning rate dropped to 0.01 after epoch 10 with no weight decay) led to the best results. We also tried training the models longer but didn't find much improvement, so we trained for the same number of epochs as in the paper (100 for COCO-Stuff, 50 for DeepFashion, 20 for AwA).

**Search for the "stage 2" models:** For "stage 2" models, we tried varying the learning rate (0.005, 0.01, 0.05, 0.1, 0.5) and found that the paper's learning rate of 0.01 produces the best results. We didn't find benefits from training the models longer, so following the original authors, we train all "stage 2" models (except *split-biased*) for 20 epochs on top of the *standard* model and use the model at the end of training as the final model. For the *CAM-based* model, we conducted an additional hyperparameter search because we got underwhelming results and degenerate CAMs with the paper's hyperparameters ($\lambda_1 = 0.1$, $\lambda_2 = 0.01$). We tried varying the regularization weight $\lambda_2$ (0.01, 0.05, 0.1, 0.5, 1.0, 5.0) and achieved the best results with $\lambda_2 = 0.1$.

# D  Selecting the best model epoch

While reproducing the *standard* model in Section 2, we tried selecting the best model epoch with four different selection methods: 1) lowest loss, 2) highest exclusive mAP, 3) highest combined exclusive and co-occur mAPs, and 4) last epoch (paper's method). Note that method 4 does not require a validation set, while methods 1-3 do as they require examinations of the loss and the mAPs at every epoch. Hence for datasets like COCO-Stuff and AwA that don't have a validation set, we can apply the first three methods only when we create a validation set by doing a random split of the original training set (e.g. 80-20 split).

In Table A4, we show COCO-Stuff *standard* results with different epoch selection methods. For methods 1–3, the best epoch is selected based on the loss or the mAPs on the validation set. For method 4, we simply select the last epoch. Note that all numbers in the table are results on the unseen test set.

First considering the model trained on the 80 split, we see that selecting the epoch with the lowest (BCE) loss yields the lowest mAP (row 1). The results of the other three methods (rows 2–4) are largely similar, with less than 0.4 mAP difference for all fields. When we plot the progression of the losses and the mAPs (Figure A1), we see that the mAPs

are mostly consistent in the latter epochs. Hence, we decided that using the last epoch is a reasonable epoch selection method. With this method we also benefit from training on the full training set, which improves all four mAPs (row 5).

Table A4: COCO-Stuff *standard* baseline results with different model epoch selection methods. All numbers are results on the test set. The best results are in bold.

| Training data | Selection method | Selected epoch | Exclusive | Co-occur | All | Non-biased |
|---|---|---|---|---|---|---|
| 80 split | 1) Lowest loss | 36 | 22.0 | 64.0 | 55.4 | 71.8 |
| 80 split | 2) Highest exclusive mAP | 79 | 22.9 | 64.1 | 55.2 | 71.6 |
| 80 split | 3) Highest exclusive + co-occur mAP | 68 | 23.0 | 64.2 | 55.3 | 71.8 |
| 80 split | 4) Last epoch | 100 | 22.9 | 63.8 | 55.0 | 71.4 |
| Full training set | 4) Last epoch | 100 | **23.9** | **65.0** | **55.7** | **72.3** |

Figure A1: Losses and mAPs of the COCO-Stuff *standard* model trained on the 80 split of the original training set. The validation loss and the four mAPs are calculated on the remaining 20 split which we use as the validation set.

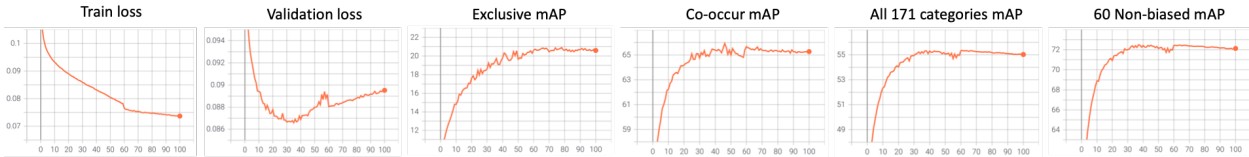

# E   Computational requirements

In Table A5, we report the single-epoch training time for each method trained with a batch size of 200 using a single RTX 3090 GPU, except for *CAM-based* which is trained on two GPUs due to memory constraints. Overall, the total training time for each method range from 35-43 hours on COCO-Stuff, 22-29 hours on DeepFashion, and 7-8 hours on AwA. For inference, a single image forward pass takes 9.5ms on a single RTX 3090 GPU. Doing inference on the entire test with a batch size of 100 takes 5.6 minutes for COCO-Stuff (40,504 images), 2.7 minutes for DeepFashion (40,000 images), 1.8 minutes for AwA (6,985 images), and 18.2 seconds for UnRel (1,071 images).

Table A5: Single-epoch training time (in minutes) for different methods, trained using a batch size of 200.

| Method | COCO-Stuff | DeepFashion | AwA |
|---|---|---|---|
| *standard* | 12.9 | 16.8 | 8.8 |
| *remove labels* | 12.8 | 16.8 | 8.8 |
| *remove images* | 8.4 | 16.1 | 0.5 |
| *split-biased* | 12.9 | 16.7 | 8.8 |
| *weighted* | 12.9 | 16.8 | 8.8 |
| *negative penalty* | 12.8 | 16.8 | 8.8 |
| *class-balancing* | 12.8 | 16.9 | 8.8 |
| *attribute decorrelation* | - | - | 12.8 |
| *CAM-based* | 17.3 | - | - |
| *feature-split* | 13.3 | 20.9 | 10.0 |

# F   Additional results

In Figure A2, we show visual comparison of our results and the paper's results reported in Table 2 for the AwA and DeepFashion datasets. A similar plot for COCO-Stuff is presented in Figure 2.

# G   Additional qualitative analyses

In Figures 6 through 9 of the original paper, the CAMs produced by the *CAM-based* and *feature-split* methods are compared to those of the *standard* model. Since the image IDs of the images used in these figures were not made available, we attempted to find images that closely replicated those used in the paper.

Figure A2: Performance of different methods on DeepFashion and AwA. The blue and red lines mark the paper's and our *standard* mAPs. All results can be found in Table 2.

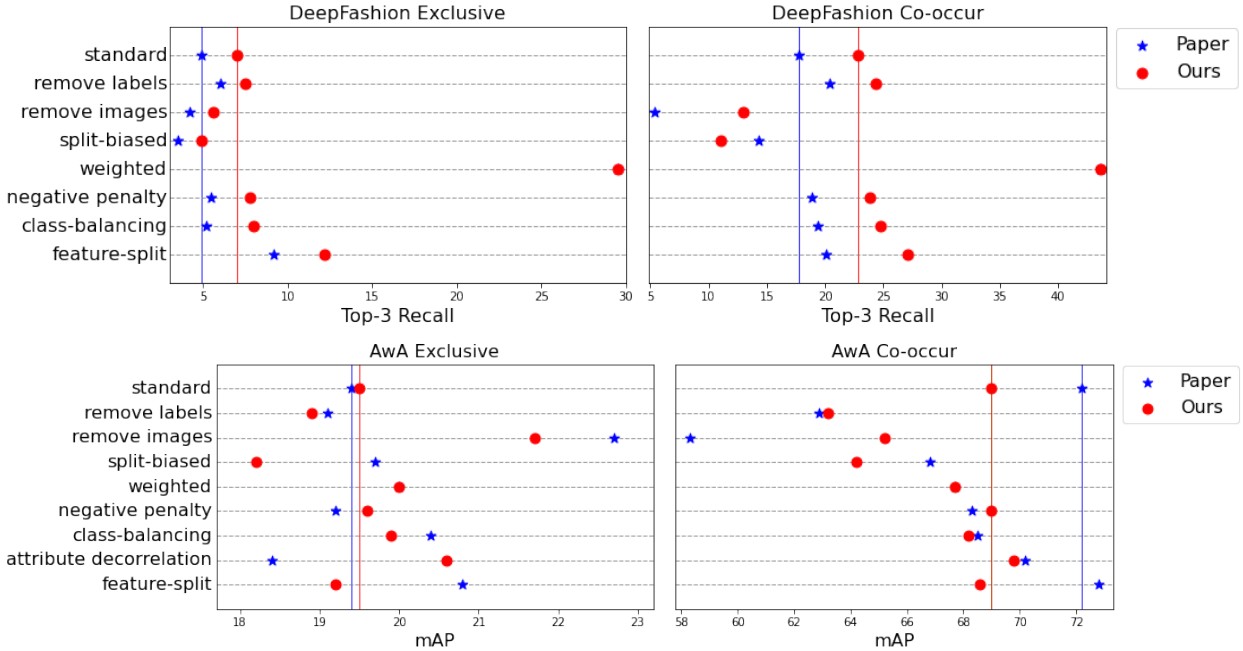

Figures 6 and 7 of the original paper compare the CAMs of the *CAM-based* method against those of the *standard* and *feature-split* method. The paper's comparison between the *CAM-based* and *feature-split* models shows that the *feature-split* CAM regions cover both $b$ and $c$ categories, whereas the *CAM-based* model's CAM covers mostly the area of $b$. In the majority of our examples, we found that this distinction to be less clear (see Figure A4). Likewise, the CAMs of our *CAM-based* method compared to the CAMs of our *standard* model are also only slightly different, even on instances where the *CAM-based* model succeeds but the *standard* model fails (see Figure A3).

Figure 8 in the original paper gives several examples images in which biased categories $b$ appear away from their context $c$. Specifically, there are examples for which the *feature-split* model was able to predict $b$ correctly but the *standard* model failed to do so, as well as some examples where both models failed. Our Figure A5 shows some of our own examples. Several of the examples from the original paper also came up in our own analysis. Out of all the test images, we found 1 "skateboard" examples on which our *feature-split* model was successful but our *standard* model failed, and 11 examples on which both models failed. There were 3 "microwave" examples on which only *feature-split* was successful and 131 examples on which neither model was successful. For "snowboard", there were 4 examples on which only the *feature-split* model was successful and 4 examples on which both failed.

Figure 9 of the original paper shows how the CAMs derived from $W_o$ and $W_s$, the two halves of the *feature-split* model's feature subspace, focus on the object $b$ and the context $c$, respectively. In our qualitative observations shown in Figure A6, we noticed the same trend.

skateboard                     remote

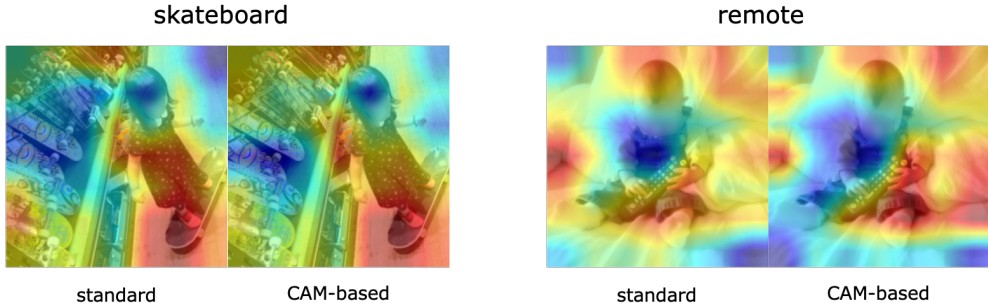

standard        CAM-based          standard        CAM-based

Figure A3: CAMs of examples on which our *CAM-based* model succeeds and our *standard* model fails. They are visually quite similar.

skateboard                     skis

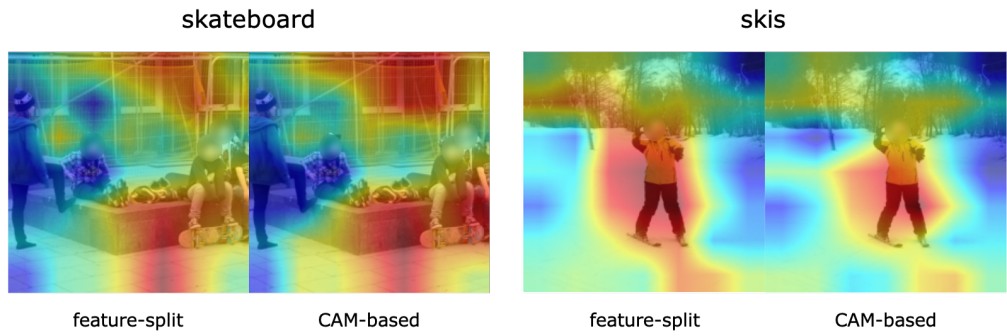

feature-split      CAM-based          feature-split      CAM-based

Figure A4: CAMs of examples on which our *feature-split* model succeeds and our *CAM-based* model fails. They are visually quite similar.

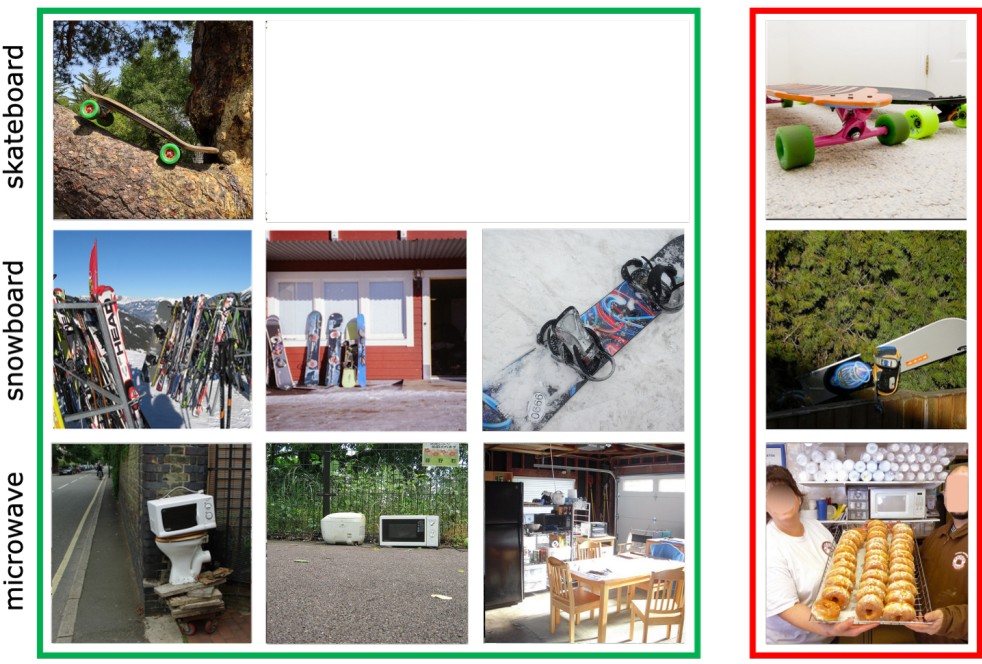

Figure A5: Examples on which our *feature-split* model succeeds and our *standard* model fails are outlined in green (left box). Examples on which both models fail are outlined in red (right box). While the original paper shows three examples of images containing *skateboard* on which the *feature-split* model succeeds but the *CAM-based* model fails, we only found one.

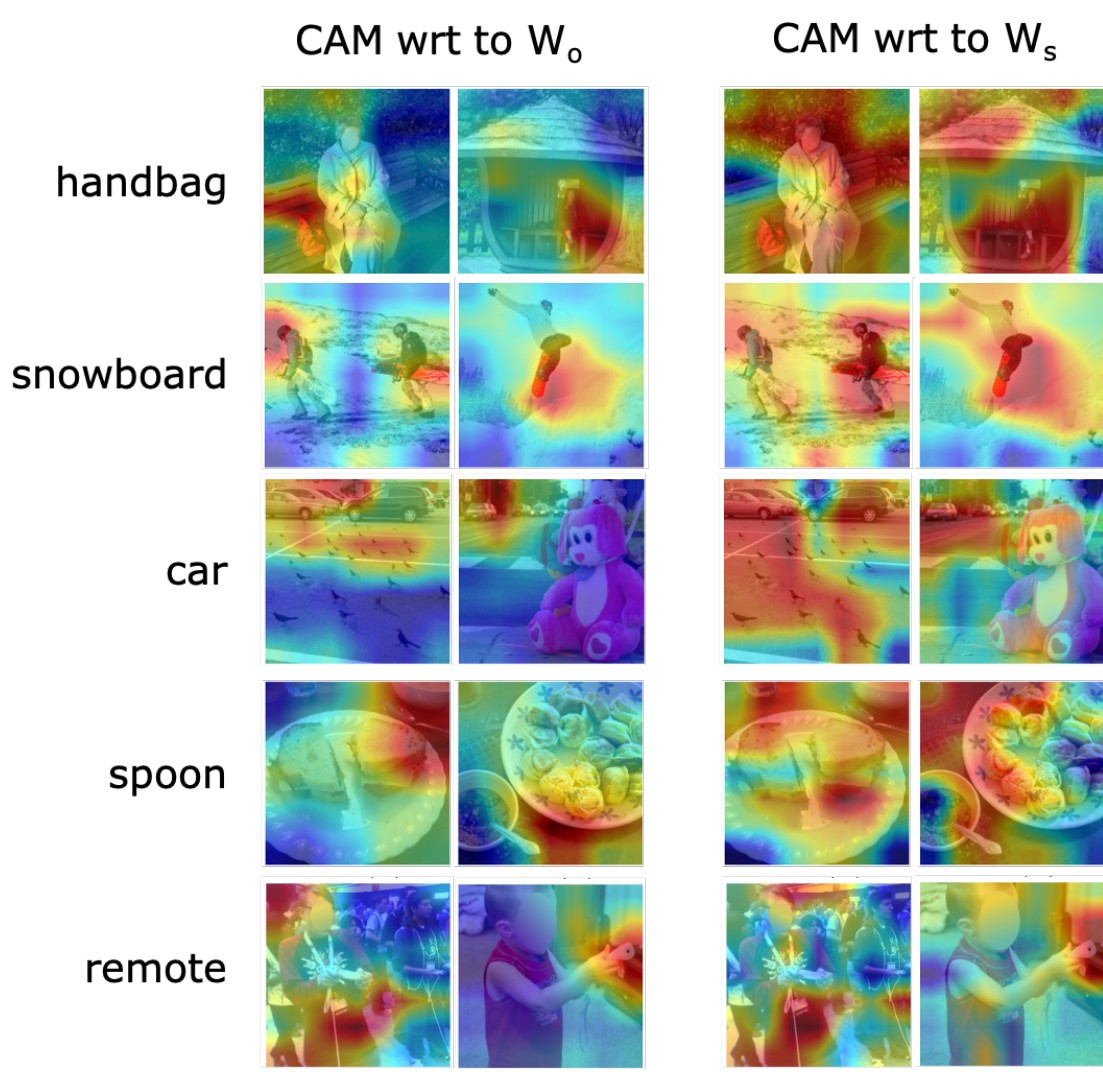

Figure A6: Interpreting the *feature-split* method by visualizing the CAMs with respect to $W_o$ and $W_s$. Consistent with the paper's observations, we see that $W_o$ focuses on the actual category (e.g., handbag, snowboard, car, spoon, remote) while $W_s$ looks at context (e.g., person, road, bowl).

# H Per-category results

In Table 2, we reported results aggregated over multiple categories. In this section, we present per-category results for the *standard*, *CAM-based*, and *feature-split* methods in Tables A6 (COCO-Stuff), A7 (DeepFashion), and A8 (AwA), and compare them to the paper's results. We also present our results on the UnRel dataset in Table A9.

Table A6: Per-category results on **COCO-Stuff**. This table together with Table A1 reproduce the paper's Table 10.

| Metric: mAP | | Exclusive | | | | | | Co-occur | | | | |
| --- | --- | --- | --- | --- | --- | --- | --- | --- | --- | --- | --- | --- |
| Biased category pairs | | *standard* | | *CAM-based* | | *feature-split* | | *standard* | | *CAM-based* | | *feature-split* |
| Biased ($b$) | Context ($c$) | Paper | Ours | Paper | Ours | Paper | Ours | Paper | Ours | Paper | Ours | Paper | Ours |
| cup | dining table | 33.0 | 29.5 | 35.4 | 30.9 | 27.4 | 23.2 | 68.1 | 61.7 | 63.0 | 59.2 | 70.2 | 63.7 |
| wine glass | person | 35.0 | 34.8 | 36.3 | 38.3 | 35.1 | 36.3 | 57.9 | 55.9 | 57.4 | 54.0 | 57.3 | 55.4 |
| handbag | person | 3.8 | 2.8 | 5.1 | 3.8 | 4.0 | 2.8 | 42.8 | 40.6 | 41.4 | 40.3 | 42.7 | 41.0 |
| apple | fruit | 29.2 | 24.6 | 29.8 | 25.5 | 30.7 | 25.6 | 64.7 | 65.6 | 64.4 | 65.0 | 64.1 | 62.6 |
| car | road | 36.7 | 36.4 | 38.2 | 39.2 | 36.6 | 36.5 | 79.7 | 79.1 | 78.5 | 78.0 | 79.2 | 78.7 |
| bus | road | 40.7 | 41.0 | 41.6 | 43.8 | 43.9 | 43.3 | 86.0 | 85.1 | 85.3 | 84.3 | 85.4 | 84.3 |
| potted plant | vase | 37.2 | 38.7 | 37.8 | 40.2 | 36.5 | 37.8 | 50.0 | 48.7 | 46.8 | 46.2 | 46.0 | 44.9 |
| spoon | bowl | 14.7 | 13.8 | 16.3 | 14.9 | 14.3 | 13.3 | 42.7 | 35.6 | 35.9 | 33.3 | 42.6 | 36.3 |
| microwave | oven | 35.3 | 41.0 | 36.6 | 43.4 | 39.1 | 41.8 | 60.9 | 60.2 | 60.1 | 59.5 | 59.6 | 59.3 |
| keyboard | mouse | 44.6 | 44.3 | 42.9 | 46.9 | 47.1 | 45.2 | 85.0 | 84.4 | 83.3 | 83.9 | 85.1 | 83.8 |
| skis | person | 2.8 | 5.4 | 7.0 | 14.1 | 27.0 | 26.8 | 91.5 | 90.6 | 91.3 | 90.7 | 91.2 | 90.5 |
| clock | building | 49.6 | 49.4 | 50.5 | 50.5 | 45.5 | 43.6 | 84.5 | 84.7 | 84.7 | 84.6 | 86.4 | 86.6 |
| sports ball | person | 12.1 | 3.2 | 14.7 | 6.5 | 22.5 | 9.5 | 75.5 | 70.9 | 75.3 | 70.7 | 74.2 | 69.7 |
| remote | person | 23.7 | 22.2 | 26.9 | 24.8 | 21.2 | 20.4 | 70.5 | 70.3 | 67.4 | 68.1 | 72.7 | 71.4 |
| snowboard | person | 2.1 | 5.0 | 2.4 | 11.6 | 6.5 | 12.7 | 73.0 | 75.6 | 72.7 | 75.7 | 72.6 | 74.9 |
| toaster | ceiling | 7.6 | 6.4 | 7.7 | 6.5 | 6.4 | 6.2 | 5.0 | 6.1 | 5.0 | 5.0 | 4.4 | 5.1 |
| hair drier | towel | 1.5 | 1.3 | 1.3 | 1.3 | 1.7 | 1.5 | 6.2 | 7.6 | 6.2 | 7.7 | 6.9 | 11.4 |
| tennis racket | person | 53.5 | 55.1 | 59.7 | 58.5 | 61.7 | 61.6 | 97.6 | 97.4 | 97.5 | 97.4 | 97.5 | 97.3 |
| skateboard | person | 14.8 | 21.1 | 22.6 | 30.5 | 34.4 | 42.0 | 91.3 | 91.7 | 91.1 | 91.7 | 90.8 | 91.1 |
| baseball glove | person | 12.3 | 2.2 | 14.4 | 7.2 | 34.0 | 31.7 | 91.0 | 88.9 | 91.3 | 89.0 | 91.1 | 88.6 |
| Mean | - | 24.5 | 23.9 | 26.4 | 26.9 | 28.8 | 28.1 | 66.2 | 65.0 | 64.9 | 64.2 | 66.0 | 64.8 |

Table A7: Per-category results on **DeepFashion**. This table together with Table A2 reproduce the paper's Table 11.

| Metric: top-3 recall | | Exclusive | | | | Co-occur | | | |
| --- | --- | --- | --- | --- | --- | --- | --- | --- | --- |
| Biased category pairs | | *standard* | | *feature-split* | | *standard* | | *feature-split* | |
| Biased ($b$) | Context ($c$) | Paper | Ours | Paper | Ours | Paper | Ours | Paper | Ours |
| bell | lace | 5.4 | 14.1 | 22.8 | 21.7 | 3.1 | 9.4 | 9.4 | 15.6 |
| cut | bodycon | 8.6 | 10.9 | 12.5 | 15.2 | 29.3 | 37.9 | 36.2 | 44.8 |
| animal | print | 0.0 | 0.0 | 1.9 | 11.5 | 1.9 | 1.9 | 2.8 | 9.4 |
| flare | fit | 18.4 | 19.4 | 32.0 | 29.1 | 56.0 | 41.9 | 62.0 | 56.2 |
| embroidery | crochet | 4.1 | 5.4 | 1.8 | 3.6 | 4.8 | 4.8 | 0.0 | 0.00 |
| suede | fringe | 12.0 | 18.5 | 19.6 | 22.8 | 65.2 | 65.2 | 73.9 | 73.9 |
| jacquard | flare | 0.0 | 0.0 | 0.9 | 6.5 | 0.0 | 9.1 | 9.1 | 18.2 |
| trapeze | striped | 8.7 | 16.5 | 29.9 | 30.7 | 42.9 | 35.7 | 50.0 | 64.3 |
| neckline | sweetheart | 0.0 | 0.6 | 0.0 | 1.3 | 0.0 | 0.0 | 0.0 | 0.0 |
| retro | chiffon | 0.0 | 0.0 | 0.4 | 1.3 | 0.0 | 0.0 | 0.0 | 0.0 |
| sweet | crochet | 0.0 | 0.0 | 0.5 | 3.7 | 0.0 | 3.5 | 0.0 | 3.5 |
| batwing | loose | 11.0 | 7.0 | 12.0 | 14.0 | 27.5 | 22.5 | 15.0 | 20.0 |
| tassel | chiffon | 13.0 | 15.3 | 16.8 | 23.7 | 25.0 | 62.5 | 25.0 | 62.5 |
| boyfriend | distressed | 11.6 | 17.7 | 11.6 | 20.0 | 49.2 | 57.1 | 38.1 | 50.8 |
| light | skinny | 2.0 | 4.0 | 1.3 | 6.4 | 14.9 | 17.0 | 8.5 | 12.8 |
| ankle | skinny | 1.0 | 7.3 | 14.6 | 11.5 | 13.2 | 35.3 | 27.9 | 32.4 |
| french | terry | 0.0 | 0.0 | 0.8 | 6.6 | 9.6 | 20.2 | 7.9 | 30.9 |
| dark | wash | 2.6 | 0.5 | 2.1 | 3.1 | 8.7 | 2.9 | 13.0 | 15.9 |
| medium | wash | 0.0 | 0.0 | 0.0 | 0.00 | 0.0 | 5.7 | 0.0 | 2.9 |
| studded | denim | 0.0 | 2.1 | 3.2 | 10.5 | 4.0 | 24.0 | 24.0 | 28.0 |
| Mean | - | 4.9 | 7.0 | 9.2 | 12.2 | 17.8 | 22.8 | 20.1 | 27.1 |

Table A8: Per-category results on **AwA**. This table together with Table A3 reproduce the paper's Table 12.

| Metric: mAP | | Exclusive | | | | Co-occur | | | |
|---|---|---|---|---|---|---|---|---|---|
| Biased category pairs | | *standard* | | *feature-split* | | *standard* | | *feature-split* | |
| Biased (b) | Context (c) | Paper | Ours | Paper | Ours | Paper | Ours | Paper | Ours |
| white | ground | 24.8 | 27.5 | 24.6 | 31.5 | 85.8 | 86.3 | 86.2 | 82.6 |
| longleg | domestic | 18.5 | 12.0 | 29.1 | 9.4 | 89.4 | 79.8 | 89.3 | 75.3 |
| forager | nestspot | 33.6 | 30.9 | 33.4 | 30.5 | 96.6 | 95.5 | 96.5 | 94.6 |
| lean | stalker | 11.5 | 12.3 | 12.0 | 10.9 | 54.5 | 51.9 | 55.8 | 55.4 |
| fish | timid | 60.2 | 54.6 | 57.4 | 54.4 | 98.3 | 97.8 | 98.3 | 97.8 |
| hunter | big | 4.1 | 3.4 | 3.6 | 3.2 | 32.9 | 34.8 | 30.0 | 42.4 |
| plains | stalker | 6.4 | 13.4 | 6.0 | 7.6 | 44.7 | 39.8 | 59.9 | 55.3 |
| nocturnal | white | 13.3 | 12.0 | 13.1 | 13.2 | 71.2 | 55.5 | 60.5 | 48.7 |
| nestspot | meatteeth | 13.4 | 14.3 | 14.9 | 15.0 | 62.8 | 62.1 | 67.6 | 57.1 |
| jungle | muscle | 33.3 | 30.4 | 31.3 | 32.2 | 88.6 | 86.3 | 86.6 | 86.7 |
| muscle | black | 9.3 | 10.1 | 9.3 | 10.0 | 76.6 | 79.3 | 73.6 | 81.5 |
| meat | fish | 4.5 | 3.7 | 3.8 | 3.3 | 76.1 | 67.7 | 73.6 | 65.0 |
| mountains | paws | 10.9 | 9.8 | 10.0 | 8.3 | 49.9 | 51.6 | 39.9 | 48.5 |
| tree | tail | 36.5 | 42.7 | 55.0 | 41.1 | 93.2 | 93.8 | 92.7 | 91.4 |
| domestic | inactive | 11.9 | 13.1 | 13.1 | 13.2 | 73.7 | 71.7 | 76.6 | 75.2 |
| spots | longleg | 43.8 | 46.9 | 45.2 | 49.7 | 61.8 | 42.6 | 59.1 | 39.3 |
| bush | meat | 19.8 | 20.1 | 22.1 | 19.7 | 70.2 | 43.1 | 75.1 | 41.7 |
| buckteeth | smelly | 7.8 | 9.1 | 8.9 | 9.3 | 27.1 | 49.1 | 45.3 | 40.0 |
| slow | strong | 15.5 | 15.0 | 14.6 | 15.0 | 95.8 | 96.4 | 93.3 | 96.6 |
| blue | coastal | 8.4 | 8.2 | 8.2 | 7.6 | 94.2 | 94.8 | 95.8 | 97.0 |
| Mean | - | 19.4 | 19.5 | 20.8 | 19.3 | 72.2 | 69.0 | 72.8 | 68.6 |

Table A9: Per-category mAP results on **UnRel**. The paper doesn't report per-category results, so we only report ours. Next to the category names are the numbers of images (out of 1,071) in which the category appears.

| Method | car (198) | bus (11) | skateboard (12) | Mean |
|---|---|---|---|---|
| *standard* | 70.0 | 44.4 | 14.5 | 43.0 |
| *remove labels* | 70.6 | 42.2 | 15.2 | 42.7 |
| *remove images* | 71.6 | 50.0 | 24.3 | 48.6 |
| *split-biased* | 60.8 | 25.9 | 0.9 | 29.2 |
| *weighted* | 71.8 | 39.5 | 22.0 | 44.4 |
| *negative penalty* | 70.6 | 42.0 | 15.0 | 42.5 |
| *class-balancing* | 70.6 | 40.7 | 15.5 | 42.3 |
| *CAM-based* | 72.0 | 40.2 | 28.2 | 46.8 |
| *feature-split* | 70.8 | 42.2 | 36.7 | 49.9 |

# I Reproducibility plan

For reference, we provide the reproducibility plan we wrote at the beginning of the project. Writing this plan allowed us to define concrete steps for reproducing the experiments and understand non-explicit dependencies within the paper. We suggest putting together a similar plan as the order in which materials are presented in the paper can be different from the order in which experiments should be run.

### Reproducibility plan

The original paper points out the dangers of contextual bias and aims to accurately recognize a category in the absence of its context, without compromising on performance when it co-occurs with context. The authors propose two methods towards this goal: (1) a method that minimizes the overlap between the class activation maps (CAM) of the co-occurring categories and (2) a method that learns feature representations that decorrelate context from category. The authors apply their methods on two tasks (object and attribute classification) and four datasets (COCO-Stuff, DeepFashion, Animals with Attributes, UnRel) and report significant boosts over strong baselines for the hard cases where a category occurs away from its typical context.

As of October 20th, 2020, the authors' code is not publicly available, so we plan to re-implement the entire pipeline. Specifically, we would like to reproduce the paper in the following order:

1. *Data preparation*: We will download the four datasets and do necessary processing.
2. *Biased categories identification*: The original paper finds a set of K=20 category pairs that suffer from contextual bias. We would like to confirm that we identify the same biased categories in COCO if we follow the process described in Section 3.1. and Section 7 in the Appendix.
3. *Baseline*: We will train the standard classifier (baseline) by fine-tuning a pre-trained ResNet-50 on all categories of COCO. The authors describe this part as stage 1 training.
4. *CAM-based method*: We will implement the proposed method which uses CAM for weak local annotation. Then using the standard classifier as the starting point, we will do stage 2 training with this method and check whether it outperforms the standard classifier.
5. *Feature splitting method*: We will implement the proposed method which aims to decouple representations of a category from its content. Then we will do stage 2 training with this method and check whether it outperforms the standard classifier and the CAM-based method.
6. *Qualitative analysis*: Once we have trained standard, ours-CAM, and ours-feature-split classifiers, we can re-create visualizations in Figures 6-9 using CAM as a visualization tool. We will compare our visualizations with the figures in the paper.

Successfully finishing 1-6 will reproduce the main claim of the paper. Afterwards, we plan to reproduce the remaining parts of the paper as time permits.

7. *Strong baselines*: In addition to the baseline standard classifier, the authors compare their two proposed methods to the following strong baselines: class balancing loss, remove co-occur labels, remove co-occur images, weighted loss, and negative penalty. With these additional baselines, we will be able to reproduce Table 2 in full.
8. *Cross dataset experiment on UnRel*: The authors test the models trained on COCO on 3 categories of UnRel that overlap with the 20 biased categories of COCO-Stuff. This experiment should be straightforward to run once the UnRel dataset is ready.
9. *Attribute classification on DeepFashion and Animals with Attributes*: To reproduce attribute classification experiments, we will compare performance of standard, class balancing loss, attribute decorrelation, and ours-feature-split classifiers on DeepFashion and Animals with Attributes datasets.

