# OpenReview forum: "[Re] Don't Judge an Object by Its Context: Learning to Overcome Contextual Bias"
_ML_Reproducibility_Challenge/2020 — RC2020_

### Official Review · AnonReviewer3 · 2021-02-27
**A detailed report with intensive experiments**

**Rating:** 7
**Confidence:** 4

**Review:**

*Problem statement:
The paper clearly states the reproducing details, together with the detailed results and difficulties.

*Presentation:
The paper is well-organized and well-written.

*Communication with original authors
The authors had some communication with the original authors.

*Code:
The code is well-organized and can be reproduced. I have tested the code on my side and all components work well.

*Recommendations for reproducibility
The authors provided useful comments for reproducing the original paper. I have read the code and found those comments are consistent with the provided codes.

*A few concerns
**The tense is not consistent in the whole paper. In some sections, the author used past tense while the present tense is used in some other sections.
** It will be even better if the authors can provide a simple illustration on the two major algorithms, like Fig.3 and Fig.4 in the original paper. The figures and simple explanations would help readers to follow your report.

**Familiar With The Original Paper:**

I have read the original paper

**Reproducibility Summary:**

Report has summary

---

### Official Review · AnonReviewer2 · 2021-02-27
**In-depth and balanced reproducibility study of great value.**

**Rating:** 9
**Confidence:** 3

**Review:**

The authors developed code for the original paper from scratch and communicated with the original authors for detail. They provide a in-depth, easily readable, well organized report and analysis. They include extensive HPO. They give valuable recommendations for future work.

Unfortunately, code will only be submitted only after the review process.

**Familiar With The Original Paper:**

I have not read the original paper

**Reproducibility Summary:**

Report has summary

---

### Author Response · Authors · 2021-03-22
**Summary of updates in the revision**

We sincerely thank the reviewers for their helpful feedback and for appreciating our reproducibility report. We’ve addressed the reviewers’ comments in the individual responses. Here we summarize additional updates we made to our report, based on further communication with the original authors after the submission.

**Updated feature-split implementation and results**. We updated our feature-split implementation, based on feedback from the original authors. Our new results are closer to the paper’s results for COCO-Stuff and DeepFashion, while lower for AwA. However, AwA results improve when we use a bigger size of $\mathrm{x}_o$. Thus, we conclude as before that the proposed feature-split methods help mitigate contextual bias in object and attribute recognition. We’ve updated all relevant tables and figures accordingly.

**COCO-Stuff biased category pairs**. We previously suspected that the original authors restricted their COCO-Stuff biased categories to be among the 80 object/thing categories. We verified with the original authors that this is indeed the case. Now out of the top-20 biased categories we identified, 18 overlap with the original paper’s, much higher than 3 we previously reported.

**DeepFashion biased category pairs**. The original authors said they performed additional cleaning of the DeepFashion biased category pairs by removing some of the overlapping and non-unique categories. To reproduce their results, we first identified the top-40 biased category pairs, then performed similar manual cleaning. 10 of our top-20 categories match with the original authors’, an increase from the 7 we previously reported.

**Additional results on all categories**. For COCO-Stuff, DeepFashion, and AwA, we added results on all categories because we found them important complementary information when discussing the relative strengths and weaknesses of different methods.

---

### Decision · Program_Chairs · 2021-03-31

**Decision:**

Accept

**Comment:**

Selected for ReScience-C Journal Publication.